# Quantifying Emotional Specificity and Ambiguity in Emojis:
# An Entropy Based Analysis of Discrete Emotion Ratings

## Abstract

Emojis are ubiquitous in digital communication, yet their emotional meanings are often ambiguous. A recently released normative data set provides mean ratings of 112 emojis on 13 discrete emotions in Spanish speakers. Based on studies demonstrating that many emoji do not unambiguously depict a single emotion, we introduce an entropy-based *emotion specificity index* (ESI) to quantify how concentrated the ratings of an emoji in one emotion. After baseline correction, we compute Shannons entropy across the 13 emotion ratings and normalize it by the maximum possible entropy. Low values of ESI indicate ambiguous or neutral emojis, whereas high values reflect a strong association with a single emotion. Our analyzes reveal that negative emojis show greater specificity than positive or neutral ones, that the principal component analysis recovers a valence continuum explaining nearly 73 % of the variance, and that ESI is systematically related to affective valence. We discuss applications of the ESI in marketing, health communication, and mental health monitoring and situate our findings within emerging normative datasets and crosscultural research on emoji interpretation.

## 1   Introduction

Emojis are pictorial symbols used to accompany written language on social platforms, in messaging applications and email. They enrich text by reinforcing tone, expressing emotions or substituting words, yet their interpretation is far from straightforward. Psychological experiments have demonstrated that many emoji are ambiguous because they do not symbolise a single emotion and instead require contextual cues for disambiguation (2; 7). A crowdsourcing study showed that only about 1 % of emoji are interpreted consistently across users, whereas roughly 4 % are as ambiguous as random words (3). At the same time, the concept of *emodiversity*the variety and relative abundance of emotions experienced by an individualhas been formalised using Shannons entropy (4). While emodiversity quantifies the diversity of an individuals emotional life, it has not been applied to pictorial symbols.

Normative datasets provide critical reference values for research on emotion processing. The *EmojiDis* database reports mean ratings of 112 emojis on thirteen discrete emotions in a large sample of Spanish speakers (1). The ratings exhibit the expected structure: positive emotions correlate strongly with each other and negatively with negative emotions, and principalcomponent analysis reveals a dominant valence dimension (1). Complementary datasets have appeared recently. Scheffler and Nenchev collected affective, semantic and descriptive norms for 107 face emojis in German speakers and replicated the quadratic relationship between valence and arousal and found that subjective familiarity correlates strongly with usage frequency and positively with valence and clarity

(5). The EmojiSP dataset provides norms for 1031 emojis across six dimensions and shows that positively valenced emojis are more familiar and frequently used than negative ones, again replicating a Ushaped valencearousal relationship (6).

Beyond normative data, research has examined how context, culture and individual differences shape emoji interpretation. Aldunate and colleagues found that perceived mood in ambiguous messages tends to be negative regardless of emoticon valence and that negative mood perception is especially pronounced when positive emoticons accompany negative text; response times are slower for incongruent messages, indicating that emoticon valence interacts with message valence during disambiguation (7). Chen et al. investigated how gender, age and culture influence emoji comprehension and reported that United Kingdom participants were more accurate than Chinese participants for most emotions except disgust; cultural differences were only partly mediated by familiarity, and Chinese participants sometimes use the smile emoji sarcastically (8). Dynamic or animated emojis introduce additional dimensions: a recent Frontiers study showed that rhythmic motion increases arousal for all dynamic emojis and that motion effects on valence depend on the emotion category, recommending that rhythm and motion be considered when designing animated emojis (9).

Emoji usage also depends on personality traits and social context. Liu and Sun found that shyness, neuroticism, extraversion and agreeableness correlate with different reasons for using emojis or stickers and that people adjust their frequency of use depending on the audience and conversation type (10). Kennison and colleagues analysed emoji use in Twitter posts and discovered that users who deploy the most emojis have the lowest openness to experience, while emoji use was unrelated to other Big Five traits; frequent emoji users also employed more words relating to family, positive emotion and sadness (11). In marketing, a systematic review concluded that emojis attract attention, stimulate social interaction, enhance consumers experiences and influence purchase decisions (12). And in medicine, the animated emoji scale (AES) for dental anxiety showed strong correlations with established scales and was preferred by 74.5 % of children, underscoring the potential of emojis in health assessment (16). These findings highlight the need for a quantitative measure of emotional specificity to guide the selection and design of emojis across applications.

We therefore ask: *How specific are individual emojis with respect to discrete emotions?* Using the EmojiDis dataset we introduce an entropybased emotion specificity index (ESI) and explore its relationship with affective valence. We hypothesise that emojis depicting clear negative expressions (e.g., disgust or anger) will exhibit high specificity, whereas neutral or skeptical faces will be ambiguous. We also consider how these findings inform realworld applications (Figure 1).

## 2 Methods

### 2.1 Dataset

We analysed the publicly available *EmojiDis* database (1). Each row corresponds to a unique emoji and includes its Unicode code point, category label (e.g., *facesmiling*, *faceneutralskeptical*) and mean ratings on thirteen discrete emotions: anger, disgust, fear, sadness, anxiety, happiness, awe, contentment, amusement, excitement, serenity, relief and pleasure. Ratings were obtained from 763 Spanish speakers on a 15 Likert scale. We subtracted 1 from each mean rating to treat the lower bound as a neutral baseline and clipped negative values to zero.

### 2.2 Emotion specificity index

Let $r_i$ denote the baselineadjusted mean rating of an emoji on emotion $i$, with $i \in \{1, \ldots, 13\}$. We define the probability $p_i = r_i / \sum_j r_j$ over all nonzero adjusted ratings and compute Shannons entropy $H = -\sum_i p_i \log p_i$. Following emodiversity research (4), the *emotion specificity index* is

$$\text{ESI} = 1 - \frac{H}{\log 13}, \tag{1}$$

where $\log 13$ is the maximum entropy for thirteen equally likely emotions. Low values of ESI indicate that ratings are spread across many emotions (ambiguity), whereas high values indicate concentration in a single emotion.

## 2.3 Valence index and principalcomponent analysis

To situate each emoji on a positivenegative continuum we defined a *valence index* as the average of the eight positive emotions (amusement, awe, excitement, happiness, pleasure, relief, contentment, serenity) minus the average of the five negative emotions (anger, disgust, fear, anxiety, sadness). Positive values denote positive affect; negative values denote negative affect. We standardised the thirteen emotion variables and performed principalcomponent analysis (PCA) to identify latent dimensions.

## 2.4 Correlation analysis and software

Pearson correlation coefficients were computed among the thirteen emotion ratings. Analyses were carried out in Python using `pandas`, `numpy` and `scikitlearn`.

# 3 Results

## 3.1 Discreteemotion structure and valence dimension

Consistent with earlier reports (1), the correlation matrix exhibited strong positive correlations among positive emotions and strong negative correlations between positive and negative emotions (see Figure 6). PCA revealed that the first principal component explained 72.8 % of the variance and loaded positively on all positive emotions and negatively on all negative emotions. This component captures an affective valence continuum (Figures 2 and 3). The second component loaded heavily on anger and sadness and accounted for 7.4 % of the variance.

## 3.2 Emotion specificity index distribution

The ESI ranged from roughly 0.015 to 0.233 (mean 0.12). Negative emojis displayed higher specificity than positive or neutral emojis. Table 1 lists the five most specific and five most ambiguous emojis. Face vomiting (🤮) and angry face with horns (👿) exhibited high ESI values and were strongly associated with *disgust* and *anger*, respectively. Conversely, grimacing face (😬) and zippermouth face (🤐) had very low ESI and were associated with *anxiety* or *anger*, indicating high ambiguity.

Table 1: Five most specific and five most ambiguous emojis according to the emotion specificity index (ESI). High specificity values indicate concentration of ratings in a single emotion; low values indicate ambiguity. Dominant emotions correspond to the highest adjusted rating.

| Emoji | Category | ESI | Dominant emotion |
|-------|----------|-----|------------------|
| 🤮 | faceunwell | 0.23 | disgust |
| 👿 | facenegative | 0.23 | anger |
| 👺 | facecostume | 0.22 | anger |
| 😧 | faceconcerned | 0.22 | anxiety |
| 😖 | faceconcerned | 0.22 | anxiety |
| 😬 | faceconcerned | 0.02 | anxiety/anger |
| 🤐 | faceneutralskeptical | 0.03 | anxiety/anger |
| 🙊 | monkeyface | 0.04 | anxiety |
| 🤨 | faceneutralskeptical | 0.05 | disbelief |
| 🤔 | facehand | 0.05 | curiosity |

## 3.3 Relationship between ESI and valence

ESI values were negatively correlated with the valence index ($r = -0.46$), indicating that more negative emojis tend to convey specific emotions. A scatter plot of emojis in the PC1PC2 plane coloured by valence index (Figure 2) shows that negative emojis cluster on the left, whereas positive emojis cluster on the right. When the same plot is coloured by ESI (Figure 3), highspecificity emojis

appear primarily among the negative cluster, whereas ambiguous emojis span the central and positive regions.

# 4 Discussion

## 4.1 Interpreting the emotion specificity index

Our entropybased ESI provides a quantitative measure of how clearly an emoji conveys a discrete emotion. High specificity implies a concentrated emotion profile and low ambiguity. Negative emojis, especially those representing anger and disgust, exhibit high specificity. This pattern may reflect the distinct facial configurations associated with negative emotions and the stronger evolutionary pressures on recognising threats. In contrast, neutral or skeptical faces have dispersed ratings across emotions and thus convey ambiguous feelings.

The valence continuum extracted by PCA aligns with the dimensional emotion theory used in many normative datasets (5; 6). The negative correlation between ESI and valence suggests that positive emojis often serve more generic functions (e.g., signalling friendliness or politeness) rather than conveying a specific discrete emotion. This observation complements work showing that positive emojis are more frequently used and more familiar than negative ones (6).

## 4.2 Crosscultural differences and individual factors

Our analyses used data from Spanish speakers and may not generalise globally. Research on emoji comprehension across cultures reveals notable differences. Chen et al. found that UK participants were more accurate than Chinese participants in identifying most emotions and that cultural differences were not fully explained by familiarity or platform; Chinese participants often used the smile emoji for sarcasm (8). These findings imply that universal facial emotions do not necessarily translate to universal emoji meanings. Personality traits also influence emoji use. Liu and Sun reported that shyness, neuroticism, extraversion and agreeableness correlate with different reasons for using emojis and stickers, and that people adjust usage depending on conversation partners and context (10). Kennison et al. observed that heavy emoji users scored lower on openness to experience and that emoji use was related to word categories such as family and sadness (11). Such individual differences likely modulate both the perceived specificity of emojis and their selection in communication.

## 4.3 Applications

**Marketing and consumer engagement.** Businesses increasingly deploy emoji in advertising and social media to stimulate interaction and influence purchasing decisions. A recent review noted that emojis attract attention, enhance creativity and innovation in marketing messages, but ambiguous emojis may hinder comprehension and must be used judiciously (12). Empirical studies with South African Generation Z consumers showed that emojis elicit emotional responses and increase purchase intention, especially among older members of the cohort (13). Our ESI can guide marketers in selecting highspecificity positive emojis (e.g., hearteyes or party face) to evoke clear positive feelings, while avoiding neutral emojis that may be misinterpreted.

**Health communication and patientprovider interaction.** Emoji can reduce the cognitive burden of health messages and increase engagement. Lin and Luos informationdesign study emphasised that emojis should be used judiciously alongside text in health materials and noted growing applications in doctorpatient communication and psychological assessment (14). In a crosscountry survey of cancer community app users, most participants reported using emojis to express emotions and believed emojis could improve communication with healthcare providers, yet they warned that variation in emoji appearance and cultural interpretation could lead to miscommunication (15). High ESI emojis may serve as reliable icons in symptom checklists or pain scales. The animated emoji scale for dental anxiety demonstrated that motion emojis are a childfriendly and valid tool for assessing anxiety, with strong correlations to established measures and a clear preference among children (16). Dynamic emojis also elicit higher arousal than static ones, and the effects of rhythm and motion on valence vary by emotion category (9), suggesting design principles for future mHealth tools.

**Mentalhealth monitoring and mHealth.** Selfhelp apps that prompt users to log mood with emojis are emerging as low burden tools for ecological momentary assessment. Van Buren et al. found that adolescents appreciated emojibased mood tracking but emphasised the need for professional support to interpret entries and avoid misunderstandings (17). Selecting emojis with high specificity could improve the reliability of mood logs; for example, using the face vomiting emoji to denote disgust or the smiling face with hearteyes for affection. Researchers should also consider normative ratings and crosscultural differences when designing such tools.

## 5 Limitations and future work

Our analysis is constrained by the characteristics of the EmojiDis dataset. Ratings were obtained exclusively from Spanish participants and contained more women than men, limiting generalisability. Future studies should compute ESI values using normative data from diverse populations and explore crosscultural consistency. Although we subtracted a neutral baseline from ratings, alternative transformations could be considered. Furthermore, context and cooccurring text dramatically change emoji interpretation (7); incorporating textual context into specificity measures is an important direction. Finally, dynamic effects and motion should be integrated into future indices to capture the richer emotional expressiveness of animated emojis.

## 6 Conclusion

We introduced an entropybased emotion specificity index to quantify how strongly an emoji conveys a particular discrete emotion. Applied to the EmojiDis dataset, the ESI revealed that negative emojis are generally more specific than positive or neutral emojis, and that neutral faces are highly ambiguous. Together with principalcomponent and valence analyses, our results provide a quantitative foundation for selecting and designing emojis for research and realworld applications. By integrating normative data, crosscultural findings and insights from marketing, health and mHealth contexts, our study offers guidance for leveraging emoji in communication while acknowledging their limitations.

## Figures

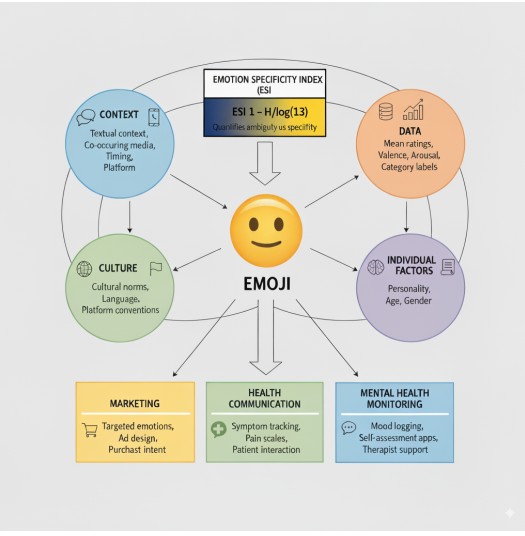

Figure 1: Conceptual model of Emoji sepcificity and its application

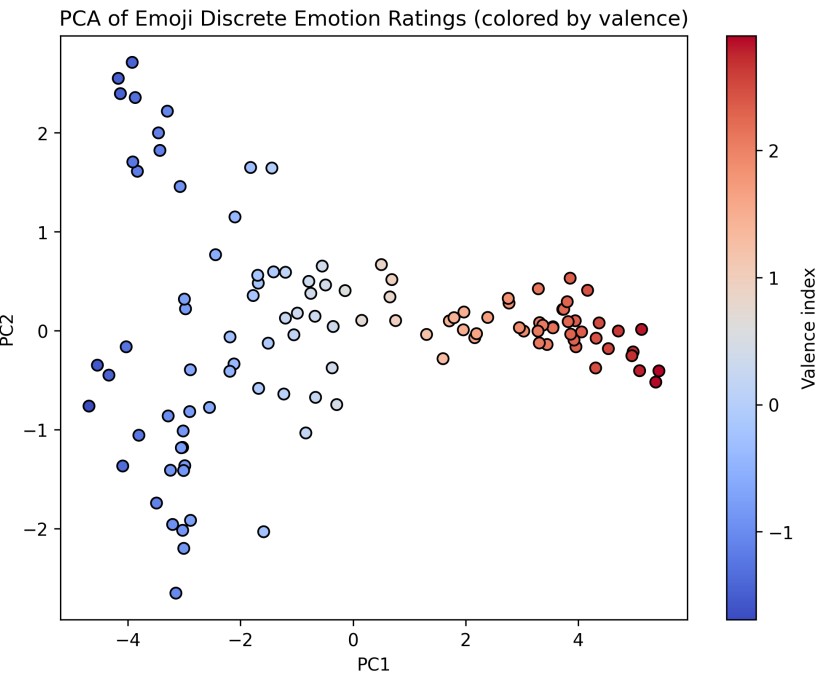

Figure 2: Principalcomponent representation of emoji ratings coloured by the valence index. Warm colours indicate positive valence and cool colours indicate negative valence. The first principal component corresponds to a valence continuum.

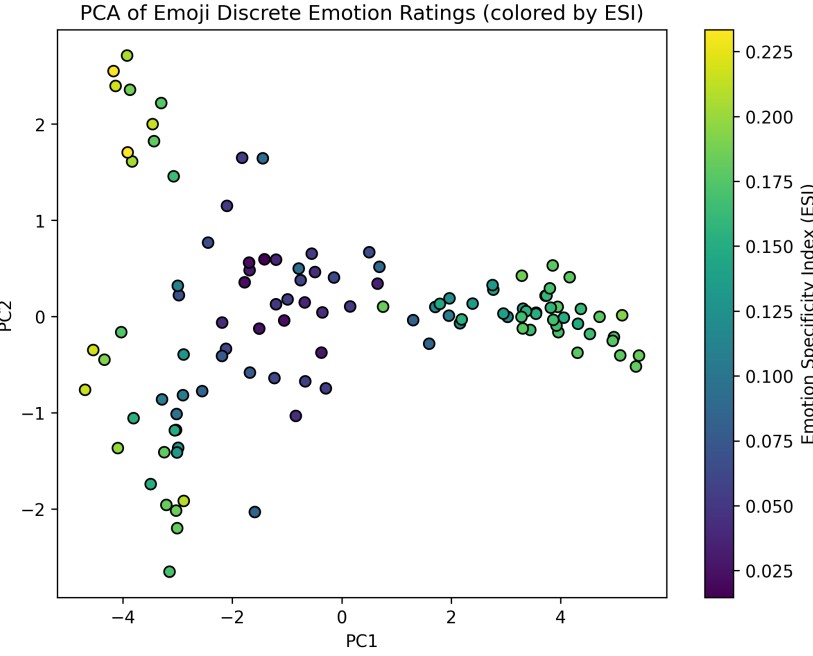

Figure 3: Principalcomponent representation coloured by the emotion specificity index (ESI). High ESI values (yellow) indicate that an emoji is strongly associated with a single emotion, while low values (dark blue) indicate ambiguity.

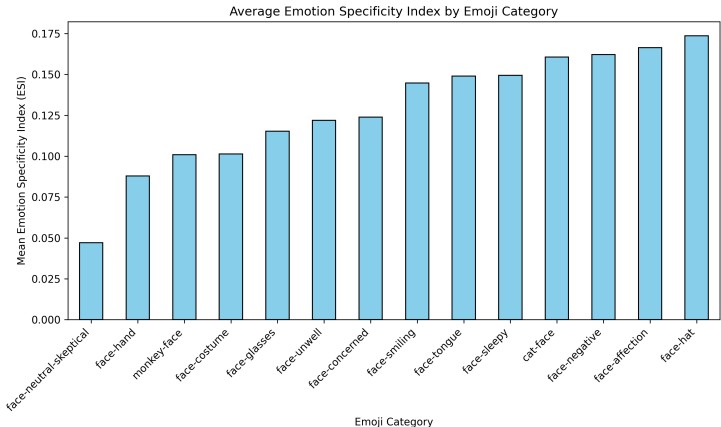

Figure 4: Mean emotion specificity index (ESI) and valence distributions across emoji categories. Categories such as *facehat* and *faceaffection* exhibit high specificity, whereas categories like *faceneutralskeptical* and *facehand* show low specificity.

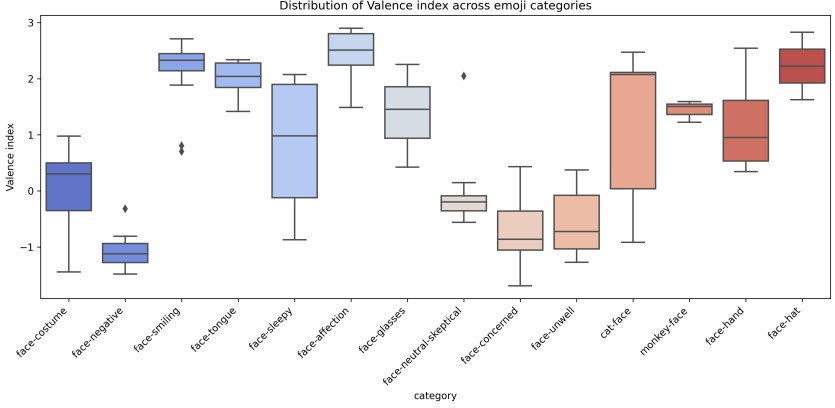

Figure 5: Distribution of the valence index across emoji categories. Positive categories show high valence and narrow spreads, while ambiguous categories show wide distributions.

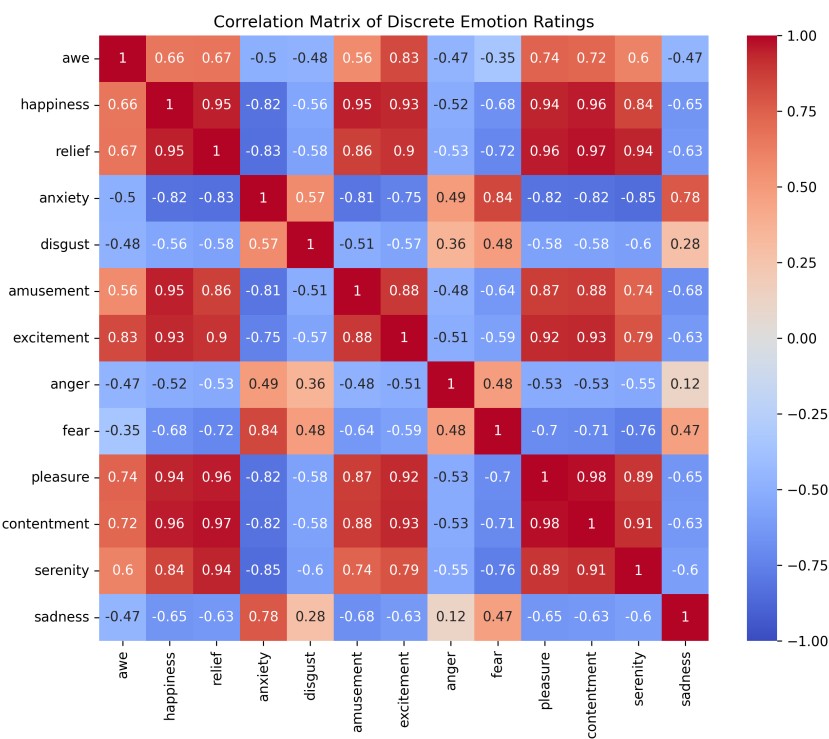

Figure 6: Correlation matrix of the thirteen discrete emotion ratings. Warm colours indicate positive correlations and cool colours indicate negative correlations. Positive emotions correlate strongly with each other and negatively with negative emotions.

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
